

# Beyond Lee-Huang-Yang description of self-bound Bose mixtures

**Miki Ota**[1⋆] **and Grigori E. Astrakharchik**[2]

**1** INO-CNR BEC Center and Dipartimento di Fisica, Università di Trento, 38123 Trento, Italy
**2** Departament de Física, Universitat Politècnica de Catalunya,
Campus Nord B4-B5, E-08034 Barcelona, Spain

⋆ miki.ota@unitn.it

## Abstract

We investigate the properties of self-bound ultradilute Bose-Bose mixtures, beyond the Lee-Huang-Yang description. Our approach is based on the determination of the beyond mean-field corrections to the phonon modes of the mixture in a self-consistent way and calculation of the associated equation of state. The newly obtained ground state energies show excellent agreement with recent quantum Monte Carlo calculations, providing a simple and accurate description of the self-bound mixtures with contact type interaction. We further show numerical results for the equilibrium properties of the finite size droplet, by adjusting the Gross-Pitaevskii equation. Our analysis is extended to the one-dimensional mixtures where an excellent agreement with quantum Monte Carlo predictions is found for the equilibrium densities. Finally, we discuss the effects of temperature on the stability of the liquid phase.



# 1 Introduction

In classical physics the formation of a liquid droplet, i.e. of a self-bound state, typically arises from the interplay between the short-range repulsive and long-range attractive components of the interatomic potential. In quantum fluids, the formation of self-bound droplets has been intensively investigated in liquid Helium, with experimental observations of strongly interacting nanodroplets in both 3He and 4He [1, 2]. Recently, it has been pointed out that an ultradilute self-bound state of matter can be formed in mixtures of ultracold atomic gases, though the underlying physics is different [3]. For binary mixtures of Bose-Einstein condensates (BECs), mean-field analysis predicts the system to become unstable against collapse when the attractive inter-species interaction overcomes the repulsive interaction between identical atoms [4]. However, in the utradilute liquid phase, the mean-field collapse is avoided as quantum fluctuations stabilize the system. The liquid droplets formed in ultracold atomic gases are fundamentally different from those in classical or Helium fluids, since they arise from beyond mean-field effects and exhibits extreme diluteness. The observation of such ultradilute liquids has been first achieved in dipolar Bose gases [5, 6], where the formation mechanism is the same, arising from an interplay between the attractive dipolar interaction and quantum fluctuations [7]. More recently, the liquid phase has been also observed in attractive Bose-Bose mixtures, both in free-space configuration [8] and confined only in one direction [9,10]. These experimental works found overall good agreement with the theory developed in the seminal work of Petrov [3].

Although the theory of Petrov [3] reckons success in explaining the stabilization mechanism and providing the energy functional, it is known that the model suffers from a serious conceptual problem. Indeed, in the relevant regime of droplet formation, the theory predicts a purely imaginary phonon velocity for the low-lying excitation spectrum, and thus a complex energy functional. In the original work [3], this problem is contoured by using the velocities calculated at the threshold point, and explicitly putting to zero the value of the suspicious phonon velocity. While such approximation is justified at the mean-field collapse point where the droplet is yet to be formed, its validity for any finite droplet is questionable. As a matter of fact, quantum Monte Carlo (QMC) method from Ref. [11] showed that the accuracy of predictions of Ref. [3] becomes worse as one increases the attractive inter-species interaction. Another issue concerns the disagreement between experiment and theory for the critical number of atoms and the droplet size, as reported in Ref. [9]. In this regard, theoretical works based on beyond mean-field variational [12] and QMC [13] approaches pointed out the crucial role played by finite-range effects. Recently, many theoretical works have been devoted to the investigation of the liquid phase in low-dimensional systems, motivated by the enhanced role of quantum fluctuations [14–16]. In particular, one-dimensional (1D) binary mixtures experimentally constitute a perfect playground due to enhanced stability, as the three-body recombination rate is greatly suppressed, and accessibility of a wide regime of interactions, as the coupling constant can even take infinite values without compromising the stability of the system. Also theoretically, 1D geometry is appealing since the energy functional does not suffer from the aforementioned imaginary part and pseudopotential interaction can be used in QMC simulations.

The aim of this paper is to provide a description of the symmetric droplet in binary mixtures of bosons, going beyond the Lee-Huang-Yang (LHY) framework. This is achieved in a phenomenological way, by explicitly including higher order corrections to the Bogoliubov speed of sound in the LHY energy. Although calculated in an approximated way, the resulting beyond LHY correction to the equation of state is found to deeply modify the equilibrium properties of the symmetric mixture. In particular, we find a strong dependence of the equilibrium density on the value of interactions, in excellent agreement with available QMC simulations.

We further investigate the equilibrium properties of the finite-size droplet within the local density approximation, and extend our analysis to the 1D mixtures. The existence of well-defined phonon modes further allow for the thermodynamic description of the self-bound state at finite temperatures. By means of phonon thermodynamics, we show that the liquid evaporates when temperature becomes comparable to the ground-state energy of the mixture.

This paper is organized as follows: in Sec. 2 we introduce the beyond LHY theory for the droplet, based on the calculation of second order terms in the long wavelength modes of the excitation spectrum. In Sec. 3 we report results obtained in the thermodynamic limit $N \to \infty$ and $V \to \infty$ with $N/V = \text{const}$. These results are compared with available QMC calculations. Section 4 is devoted to the numerical analysis of the finite-size droplet, using a generalized Gross-Pitaevskii equation. Extension of the analysis to the one-dimensional mixture is discussed in Sec. 5. In the last part of this work Sec. 6, we discuss the effects of temperature on the stability of the liquid phase.

## 2 Theory

We consider a uniform binary mixture of bosons with equal masses ($m_1 = m_2 = m$). In terms of the single-particle creation and annihilation operators in each component, $\hat{a}_{i,\mathbf{k}}^{\dagger}$ and $\hat{a}_{i,\mathbf{k}}$ ($i = 1, 2$), the Hamiltonian including all two-body collisions takes the form:

$$H = \sum_{i,\mathbf{k}} \varepsilon_{\mathbf{k}} \hat{a}_{i,\mathbf{k}}^{\dagger} \hat{a}_{i,\mathbf{k}} + \frac{1}{2V} \sum_{i,\mathbf{k},\mathbf{k}',\mathbf{q}} g_{ii} \hat{a}_{i,\mathbf{k}}^{\dagger} \hat{a}_{i,\mathbf{k}'+\mathbf{q}}^{\dagger} \hat{a}_{i,\mathbf{k}'} \hat{a}_{i,\mathbf{k}+\mathbf{q}} + \frac{g_{12}}{V} \sum_{\mathbf{k},\mathbf{k}',\mathbf{q}} \hat{a}_{1,\mathbf{k}}^{\dagger} \hat{a}_{1,\mathbf{k}+\mathbf{q}} \hat{a}_{2,\mathbf{k}'+\mathbf{q}}^{\dagger} \hat{a}_{2,\mathbf{k}'}, \quad (1)$$

where $\varepsilon_{\mathbf{k}} = \hbar^2 k^2/(2m)$ and we have assumed a contact-type interactions between particles characterized by coupling constants $g_{ij}$, related to the $s$-wave scattering length $a_{ij}$ by $g_{ij} = 4\pi\hbar^2 a_{ij}/m$. The ground state energy of the system is obtained by diagonalizing the Hamiltonian (1). This is achieved by applying the Bogoliubov prescription and replacing $\hat{a}_{i,\mathbf{k}}$ and $\hat{a}_{i,\mathbf{k}}^{\dagger}$ by the total number of atoms in each component $\sqrt{N_i}$, as well as appropriate canonical transformations. The details of the calculation can be found elsewhere [17] leading to the following form:

$$H = E + \sum_{\mathbf{k} \neq 0} \left( E_{d,\mathbf{k}} \hat{\alpha}_{\mathbf{k}}^{\dagger} \hat{\alpha}_{\mathbf{k}} + E_{s,\mathbf{k}} \hat{\beta}_{\mathbf{k}}^{\dagger} \hat{\beta}_{\mathbf{k}} \right), \quad (2)$$

where $\hat{\alpha}_{\mathbf{k}}^{\dagger}$ and $\hat{\beta}_{\mathbf{k}}^{\dagger}$ are the creation operators for the quasiparticles obeying Bose statistics. The excitation spectrum of the system reads $E_{d(s),\mathbf{k}} = \sqrt{\varepsilon_{\mathbf{k}}^2 + 2mc_{d(s)}^2 \varepsilon_{\mathbf{k}}}$ with $c_{d(s)}$ the sound velocities in the density ($d$) and spin ($s$) channels, defined hereafter. The ground state energy becomes

$$E = \sum_{i,j} \frac{g_{ij}}{2V} N_i N_j + \frac{1}{2} \sum_{\mathbf{k}} \left( \left[ E_d + E_s - 2\varepsilon_{\mathbf{k}} - m(c_d^2 + c_s^2) \right] \right). \quad (3)$$

The first term of Eq. (3) describes the mean-field internal energy, whereas the second one inside the bracket corresponds to the contribution from quantum fluctuations and is often referred to as the LHY term [18]. Within the Bogoliubov theory, the long wavelength modes of the excitation spectrum are given by the linear phonons. For a symmetric mixture $g_{11} = g_{22} = g$, the speed of sound is [4]

$$c_{d,B}^2 = \frac{1}{2m} \left[ g(n_1 + n_2) - \sqrt{g^2(n_1 - n_2)^2 + 4g_{12}^2 n_1 n_2} \right], \quad (4a)$$

$$c_{s,B}^2 = \frac{1}{2m} \left[ g(n_1 + n_2) + \sqrt{g^2(n_1 - n_2)^2 + 4g_{12}^2 n_1 n_2} \right], \quad (4b)$$

with $n_i = N_i/V$ the atomic density of each component. The main idea of our work is to extend the LHY description in a perturbative way, by evaluating the sound velocities beyond the Bogoliubov formula (4) in a self-consistent manner and to obtain the correction to the ground state energy Eq. (3). The calculation of higher order terms for the excitation spectrum can be achieved either microscopically, by developing the second-order Beliaev theory for the mixtures [19, 20], or by a much simpler macroscopic approach based on thermodynamic relations. Indeed, it is known for a single-component weakly interacting Bose gas that at $T = 0$, the velocity of the long wavelength phonon mode is related to the compressibility $\kappa$ as $c = \sqrt{(mn\kappa)^{-1}}$ [21, 22]. In an analogous way, one can relate the sound modes in the density and spin channels of the symmetric Bose mixtures, to the compressibility $\kappa_d$ and spin susceptibility $\kappa_s$ of the system, respectively: $c_{d(s)} = \sqrt{(mn\kappa_{d(s)})^{-1}}$, with $n = n_1 + n_2$ the total atom density. The identity between the microscopic phonon velocity and the macroscopic speed of sound is exact for the density mode, while for the spin mode an additional contribution known as the Andreev-Bashkin effect is missing [23]. However, it has been shown in Refs. [24–26] that for weak interactions, the Andreev-Bashkin drag has a negligible effect on the spin speed of sound as compared to the contribution arising from the susceptibility, and one shall therefore neglect it in this work. The compressibilities are obtained from the energy (3) according to the thermodynamic relation

$$n^2\kappa_d = \left(\frac{\partial^2 E/V}{\partial(n_1+n_2)^2}\right)^{-1}, \quad n^2\kappa_s = \left(\frac{\partial^2 E/V}{\partial(n_1-n_2)^2}\right)^{-1}. \tag{5}$$

In what follows, we evaluate both the speed of sound and the associated ground state energy for the three-dimensional (3D) mixtures. The extension of LHY theory in lower dimension follows essentially the same path and we will discuss as an example the one-dimensional (1D) mixture in the last part of this paper.

In 3D, the LHY contribution in Eq. (3) exhibits an ultraviolet divergence, arising from an approximate relation between the coupling constant $g_{ij} = 4\pi\hbar^2 a_{ij}/m$ and $s$-wave scattering length $a_{ij}$ in the first Born approximation. This is conveniently solved by a proper renormalization of the coupling constant [27]: $g_{ij} \rightarrow g_{ij}(1 + g/V \sum_{\mathbf{k}} m/(\hbar k)^2)$. Then, the momentum sum in Eq. (3) can be turned into an integral which can be performed analytically resulting in

$$\frac{E}{V} = \frac{g}{2}(n_1^2 + n_2^2) + g_{12}n_1 n_2 + \frac{8}{15\pi^2}\frac{m^4}{\hbar^3}\left(c_d^5 + c_s^5\right). \tag{6}$$

The regime of interest corresponds to repulsive intra-species interaction $g > 0$ and attractive inter-species interaction $g_{12} < 0$, with a small imbalance $|\delta g|/g \ll 1$ where $\delta g = g + g_{12}$. For a system satisfying the inequality $\delta g < 0$, the mean-field field theory would result in energy $\propto n^2$ given by the first two terms of Eq. (6) and would predict a collapse of a homogeneous state towards bright soliton formation. The beyond mean-field theory eliminates the mechanical instability as the quantum fluctuations generate a repulsive term $\propto n^{5/2}$. The interplay between such attractive and repulsive forces is at the heart of droplet formation.

However, energy (6) suffers from the presence of a dynamical instability. This can be easily seen for the unpolarized mixture ($n_1 = n_2 = n/2$), for which the Bogoliubov phonon modes (4) take the values

$$c_{d,\mathrm{B}} = \sqrt{\frac{(g+g_{12})n}{2m}}, \quad c_{s,\mathrm{B}} = \sqrt{\frac{(g-g_{12})n}{2m}}, \tag{7}$$

therefore providing a purely imaginary speed of sound for the density mode $c_d$ in the liquid phase. In the original work of Petrov [3], this issue of imaginary sound is circumvented by putting $\delta g = 0$ in Eq. (7):

$$c_{d,\mathrm{P}} = 0, \quad c_{s,\mathrm{P}} = \sqrt{\frac{gn}{m}}. \tag{8}$$

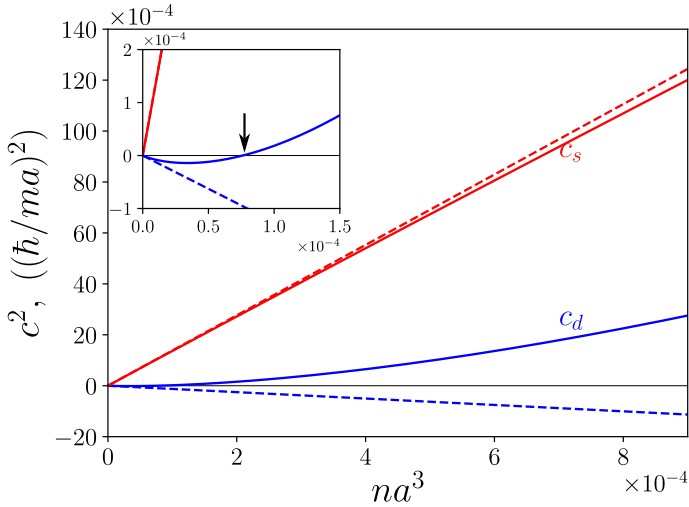

Figure 1: Square of the speed of sound as a function of the gas parameter, evaluated for the characteristic value $\delta g / g = -0.2$. The blue (bottom) and red (upper) lines are the density and spin sound velocity, respectively. Dashed lines, Bogoliubov theory, Eq. (7). Solid lines, improved theory, Eq. (11). Inset: zoom of the low-density region in the vicinity of the spinodal point, shown by the black arrow.

The imaginary phonon in the Bogoliubov theory indicates that not only the equation of state (6) needs the presence of a beyond mean-field LHY term to be stabilized, but also the sound velocity requires additional higher-order correction in order to be well defined. In the unpolarized configuration, one immediately finds from Eqs. (5)-(7) the compressibility and susceptibility of the mixture:

$$\kappa_d^{-1} = \frac{n^2}{2} \left\{ \delta g + g \sqrt{na^3} \frac{4\sqrt{2}}{\sqrt{\pi}} \left[ \left( 1 + \frac{g_{12}}{g} \right)^{5/2} + \left( 1 - \frac{g_{12}}{g} \right)^{5/2} \right] \right\}, \tag{9}$$

$$\kappa_s^{-1} = \frac{n^2}{2} (g - g_{12}) \left\{ 1 + \frac{\delta g}{g_{12}} \sqrt{na^3} \frac{8\sqrt{2}}{3\sqrt{\pi}} \left[ \left( 1 + \frac{g_{12}}{g} \right)^{3/2} - \left( 1 - \frac{g_{12}}{g} \right)^{3/2} \right] \right\}. \tag{10}$$

While compressibility and susceptibility are both complex as the non-zero imaginary part naturally arises in the perturbative approach, one notices that the imaginary component is of order $|\delta g|^{5/2}$ and can be safely neglected in respect to the real part for the experimentally relevant parameter range $|\delta g|/g \ll 1$. This is equivalent to neglecting fluctuations in the density channel, while preserving those in the spin channel. Therefore using the identity $c_{d(s)} = \sqrt{(mn\kappa_{d(s)})^{-1}}$, we obtain the following beyond Bogoliubov expressions for the speed of sound:

$$c_d^2 \simeq \frac{n}{2m} \left[ \delta g + g \sqrt{na^3} \frac{4\sqrt{2}}{\sqrt{\pi}} \left( 1 - \frac{g_{12}}{g} \right)^{5/2} \right], \tag{11a}$$

$$c_s^2 \simeq \frac{n}{2m} (g - g_{12}) \left[ 1 - \frac{\delta g}{g_{12}} \sqrt{na^3} \frac{8\sqrt{2}}{3\sqrt{\pi}} \left( 1 - \frac{g_{12}}{g} \right)^{3/2} \right]. \tag{11b}$$

We show in Fig. 1 a comparison between the Bogoliubov sound velocity Eq. (7) and higher order sound velocity (11). One striking feature which our theory predicts is that the velocity of the density mode becomes real above a certain density when beyond mean-field corrections are included, while it is a purely imaginary quantity in the Bogoliubov description. Another feature is that even in the higher order description, the speed of sound becomes imaginary

in a small window of density $na^3 \lesssim (\delta g/g)^2$ (see inset of Fig. 1). However, this instability has a physical nature and it defines the spinodal point below which the uniform liquid is unstable towards the formation of multiple droplets. That is, at zero pressure, the liquid is self-bound and it stays at the equilibrium density which corresponds to the position of the minimum in the equation of state. If positive (negative) pressure is applied, the density of the liquid increases (decreases) with respect to the equilibrium density and the energy increases. If the applied pressure is large and positive the energy eventually becomes positive, still the homogeneous system remains stable. On the contrary, for large negative pressures the homogeneous shape can no longer be sustained and the liquid fragments into droplets each having density close to the equilibrium one. Experimentally, the fragmentation instability below the spinodal point can be investigated by applying an external field exerting a large enough negative pressure on the liquid. Alternatively, the spinodal decomposition can be experimentally observed by quenching the scattering lengths in such a way that the system is brought fast from the stable to the unstable region of the phase diagram. In addition, our predictions for the speeds of sound can be verified from determination of the excitation spectrum using Bragg spectroscopy [28, 29], or by observing the propagation of sound waves upon applying density/magnetic excitation [30, 31].

Once we have shown that higher-order corrections remove the unphysical instability associated with the complex values of the speed of density mode, it is useful to investigate if the predictions for the ground-state energy can be also improved.

## 3 Energy analysis

We now recalculate the LHY term using the beyond-Bogoliubov sound velocities Eq. (11) and improve the equation of state (3). The resulting energy is shown in Fig. 2 with a red solid line. It has a shape typical for a liquid with the minimum associated with the equilibrium density. It is instructive to compare our results with the ones obtained using the original prescription from Ref. [3], Eq. (8) (black dashed line) and the LHY ground state energy calculated with the Bogoliubov sounds (7) (black dotted line). We remind that in the latter case, the energy is complex, and we only show its real part in Fig. 2. Taking as reference the LHY energy with the Bogoliubov dispersion law, one can see that inclusion of higher order terms in the density sound (11a) (top green dashed line) changes only slightly the behavior of the energy. Instead, the inclusion of higher order terms in the spin sound (11b) (blue dashed-dotted line) strongly suppresses the energy. The contributions arising from different approaches can be conveniently classified if one normalizes both the energy and the density, to their equilibrium values obtained within the approach of Petrov [3]:

$$\frac{|E_0|}{N} = \frac{25\pi^2\hbar^2|a+a_{12}|^3}{49152ma^5}, \tag{12}$$

$$n_0 = \frac{25\pi(a+a_{12})^2}{16384a^5}, \tag{13}$$

and expand the energy $E/|E_0|$ in series of the small parameter $\delta g/g$:

$$\frac{E}{|E_0|} \simeq -3\frac{n}{n_0} + 2\left(\frac{n}{n_0}\right)^{3/2} + \frac{5}{2}\frac{|\delta g|}{g}\left(\frac{n}{n_0}\right)^{3/2} - \frac{5}{4}\left(\frac{\delta g}{g}\right)^2\left(\frac{n}{n_0}\right)^{3/2}\left(\frac{5}{3}\sqrt{\frac{n}{n_0}} - \frac{3}{2}\right). \tag{14}$$

The two first terms are identified as the ones of the Petrov theory, and expressed in these units, they do not depend explicitly on $\delta g$. The third term comes from the spin sound of the Bogoliubov theory (7), and gives a positive shift of the energy, as one can verify on Fig. 2 (black

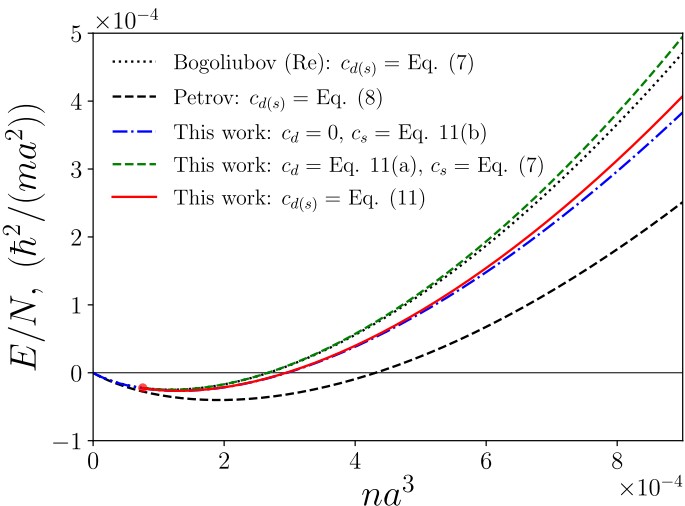

Figure 2: Energy per particle as a function of density for $\delta g/g = -0.2$. The black dotted line is the real part of the LHY result Eq. (3) with the Bogoliubov sound (7), while the black dashed line is obtained by setting $g_{12}/g = -1$ in the LHY term as it was originally done in Ref. [3]. The green dashed and blue dotted-dashed lines correspond, respectively, to the inclusion of higher order term in the density (11a) and spin (11b) sound velocity. The red solid line includes both density and spin corrections. The circle indicates the spinodal density.

dotted line). Finally, the last terms come from the corrections brought to the spin sound within our new theory (11b). It is a negative contribution in the region where $n/n_0 \gtrsim 1$, resulting in a suppression of the energy (see the blue dotted-dashed line in Fig. 2). As for the density sound (11a), the first contribution to the energy functional enters with a higher power as $(|\delta g|/g)^{5/2}$. Even though the speed of density sound is drastically modified in our theory, its effect on the energy remains therefore tiny. Thus, we conclude that the main correction to the equation of state arises from quantum fluctuations in the spin channel. It is worth noticing that the inclusion of density sound leads to a spinodal point below which the uniform liquid is unstable against density fluctuations (filled circle in Fig. 2).

Although our theory provides a higher-order correction to the speed of sound, still not all second-order terms are taken into account as it would be in Beliaev theory [20]. Thus it is important to verify the validity of our results through a direct comparison with available Monte-Carlo calculations [11]. Figure 3 shows $E/|E_0|$ calculated for different values of interaction disbalance $\delta g/g$, as a function of density $n/n_0$. As mentioned before, in the rescaled form the energies evaluated within the original Petrov theory collapse to a single curve (shown with a black solid line in Fig. 3) which is independent of the specific value of $\delta g/g$. Predictions of our theory are shown with color lines and should be confronted with QMC results taken from Ref. [11]. The agreement between our beyond LHY theory and QMC results is surprisingly good, especially in the most interesting region around the minimum of energy. This suggests that although we do not perform a systematic calculation of the third-order terms of the perturbative theory, in practice the contributions which we miss are small in the considered case of a symmetric mixture.

The agreement of our theory with QMC results is further emphasized in Fig. 4 where we show the calculated equilibrium density (corresponding to the minimum of energy in Fig. 3) as a function of $\delta g/g$. In particular, for the largest value of $|\delta g|$ our theory predicts a decrease for the equilibrium density of about 50% in respect to the prediction from Ref. [3]. For com-

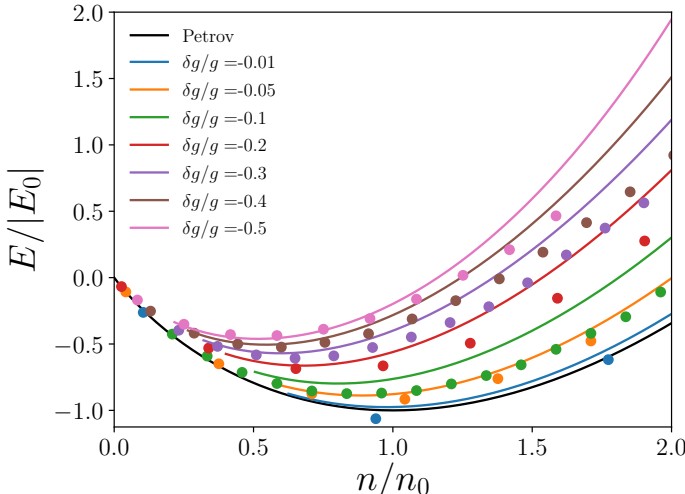

Figure 3: Equation of state in units of the equilibrium density (13) and energy (12), for different values of $\delta g/g$. The lowest black solid line is the universal result from Ref. [3]. The color solid lines are the predictions from the beyond LHY theory, while the color dots are QMC results from Ref. [11].

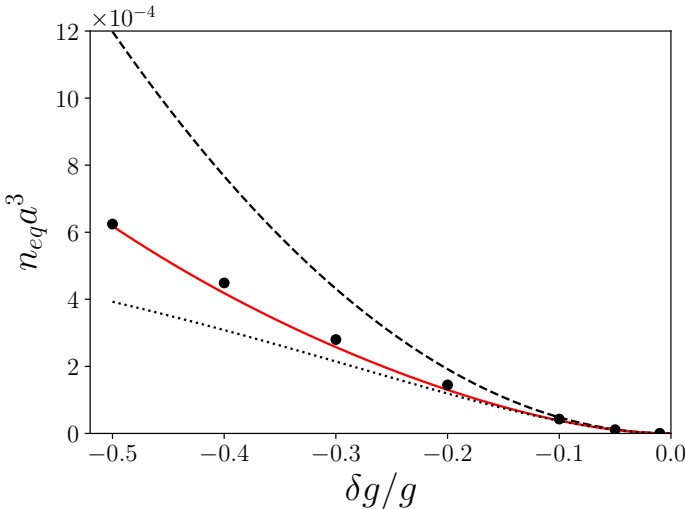

Figure 4: Equilibrium density of liquid phase as a function of $\delta g/g$. The black dashed and dotted lines correspond to speeds of sound given by Eq. (8) and Eq. (7), respectively. The red solid line is the calculation within the beyond LHY theory, relying on Eq. (11), while the black dots are QMC results from Ref. [11].

parison, we also show the equilibrium density obtained using the Bogoliubov sounds(7) in the LHY energy, which exhibits a lower value.

## 4 Finite size droplet

Following the success of the presented theory in removing the instability in the speed of sound and significantly improving the equation of state of a homogeneous liquid, we aim at an improved description of finite-size droplets. A common path to do so [3, 7, 11] is to

improve the energy functional. Differently from single-component gas, there is a separation of scales in the considered binary mixtures. That is, a finite-size droplet changes its shape at the distances of the order of the "large" healing length defined by the chemical potential, $\xi \propto \sqrt{\hbar^2/(m|\mu|)}$, while the main contributions to the LHY terms in Eq. (3) arise from "short" distances $\propto 1/c_s$ [3]. Under these specific conditions it is possible to incorporate higher-order terms locally as non-linear terms in the Gross-Pitaevskii equation (GPE) for the droplet. Following the notation of Ref. [3] we introduce the rescaled coordinate $\tilde{r} = r/\xi$ with $\xi = \sqrt{6\hbar^2/(|\delta g| m n_0)}$, where the equilibrium density $n_0$ is given in Eq. (13). Then, the energy functional associated to the equation of state (6) with the beyond Bogoliubov speed of sound (11) is written as

$$\eta \mathcal{E} = \int d\tilde{\mathbf{r}} \left\{ \frac{1}{2} |\nabla_{\tilde{\mathbf{r}}} \Phi|^2 - \frac{3}{2} |\Phi|^4 + \frac{1}{4\sqrt{2}} |\Phi|^5 \left( 1 - \frac{g_{12}}{g} \right)^{5/2} \right.$$
$$\left. \times \left[ 1 - |\Phi| \frac{5}{24\sqrt{2}} \frac{g}{|g_{12}|} \left( \frac{\delta g}{g} \right)^2 \left( 1 - \frac{g_{12}}{g} \right)^{3/2} \right]^{5/2} \right\}, \tag{15}$$

where $\eta = 6\xi^3/(n_0^2 |\delta g|)$ and the classical field $\Phi$ is normalized as $\int d\mathbf{r} |\Phi|^2 = N/n_0$. We briefly note that we consider $\Phi_1 = \Phi_2 = \Phi$, and neglect therefore the internal dynamics between the respective components [3,32]. In Eq. (15) we have assumed $c_d = 0$ to hold, so as to avoid the imaginary part of the energy functional. This is motivated from the analysis of the previous section, in which neglecting the density mode was found not to alter greatly the behavior of the equation of state. The energy functional Eq. (15) reduces to the one used in Ref. [3] in the limit $\delta g \to 0$.

The GPE can be obtained from the variational procedure $i\hbar \partial \Phi/\partial \tilde{t} = \eta \partial \mathcal{E}/\partial \Phi^*$ [4]:

$$i\hbar \frac{\partial \Phi}{\partial \tilde{t}} = \left[ -\frac{1}{2} \Delta_{\tilde{\mathbf{r}}} - 3|\Phi|^2 + \frac{5}{8\sqrt{2}} |\Phi|^3 \left( 1 - \frac{g_{12}}{g} \right)^{5/2} (1 - \alpha |\Phi|)^{3/2} \left( 1 - \frac{3}{2} \alpha |\Phi| \right) \right] \Phi, \tag{16}$$

with $\alpha = \frac{5}{24\sqrt{2}} \frac{g}{|g_{12}|} (\delta g/g)^2 (1 - g_{12}/g)^{3/2}$. In what follows, we solve numerically the stationary GPE (16) by propagating it in imaginary time [33].

Before discussing the numerical results, it is insightful to notice that the leading order correction introduced by the beyond Bogoliubov sound in the energy functional is of attractive three-body nature. Expanding the quantum fluctuations term in Eq. (15), the energy functional yields term $\mathcal{E} \propto K_3 n_0^3 |\Phi|^6/3!$. Such cubic dependence can be interpreted as corresponding to three-body interactions with the strength given by

$$\frac{K_3}{3!\hbar} \simeq -\frac{256}{9} \frac{\hbar a^4}{m} \frac{\delta a}{a_{12}} \left( 1 - \frac{a_{12}}{a} \right)^4. \tag{17}$$

An estimate using typical experimental values for $^{39}$K with $a = \sqrt{a_{11} a_{22}} \simeq 48 a_0$ where $a_0$ is the Bohr radius and $a_{12}/a = -0.115$ provides $|K_3|/3!\hbar \sim 10^{-41} \text{m}^6/\text{s}$, therefore being of the same order as the three-body loss rate measured in real experiment [8]. Thus, the effective three-body interactions (17) might be of the same order as the three-body terms which are not included in the model Hamiltonian (1). It was proposed that inclusion of three-body interactions on its own might lead to stabilization of a droplet [32,34].

We show in Fig. 5 the density profile of the self-bound mixture obtained by solving Eq. (16), in two characteristic regimes. The GP equation is governed by dimensionless parameter $\tilde{N} = N/(n_0 \xi^3)$ which is linearly proportional to the number of atoms $N$ and as well depends on the interaction strength. Once expressed in the chosen units, the density profiles of different systems with the same value of $\tilde{N}$ reduce within the approach of Ref. [3] to a single curve, shown with the top black solid line. The density profiles predicted by our theory strongly

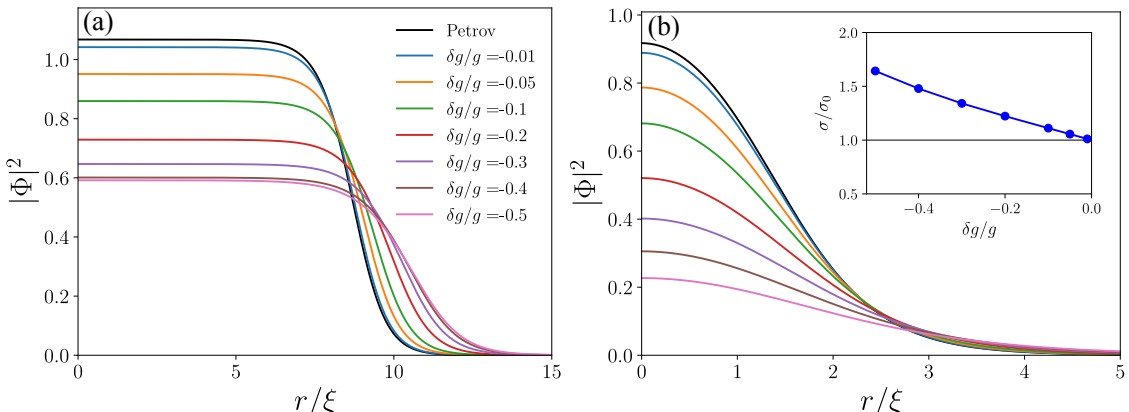

Figure 5: Density profile of the self-bound droplet obtained from the GPE (16) for (a) $N/n_0\xi^3 = 3000$ (b) $= 30$. The upper black line is the universal result of Petrov's theory [3], whereas the color lines are results from our beyond-LHY approach, calculated for different values of interaction strength. Color guides are the same for the upper and lower panel and are ordered in increasing the disbalance $\delta g/g$ from top to bottom. The inset in the lower panel reports the mean-square size of the droplet, normalized to the value $\sigma_0$ coming from Petrov's theory, as a function of $\delta g/g$.

depend on the interactions and are shown with color lines. The two characteristic examples shown in Fig. 5 are calculated for (a) $N/n_0\xi^3 = 3000$ where a bulk region is formed in the center which is a hallmark of a liquid, and (b) $N/n_0\xi^3 = 30$ corresponding to typical experimental conditions [8]. The most crucial effect is that the central density of the droplet is decreased while its size is simultaneously increased, in agreement with diminishing equilibrium density found in a homogeneous liquid, see Fig. 4. It is interesting to note that also in the experiment of Ref. [9], the measured data for the droplet size was found to be larger than the prediction from Ref. [3]. For a sufficiently small number of atoms, the density profile of the liquid phase is well described by a Gaussian function, and one can extract the width of the droplet from a fitting. The obtained result is shown in the inset of Fig. 5(b), where the size of the self-bound mixture is found to be systematically larger than the value $\sigma_0$ predicted from Petrov's theory. At the same time, $\sigma_0$ increases when $\delta g \to 0$, so the actual droplet size is larger for a smaller disbalance.

Another experimentally relevant quantity is the critical atoms number for the droplet formation. Below a certain number of atoms, the droplet state becomes unstable and it evaporates. The critical number of atoms for the unstable phase can be conveniently investigated by means of variational approach. Close to the critical number, one can indeed safely use the Gaussian ansatz and assume the density profile of the gas to be [32]

$$\Phi = \sqrt{\frac{\tilde{N}}{n_0 \pi^{3/2} \sigma^{3/2}}} \exp\left(-\frac{1}{2}\frac{r^2}{\sigma^2}\right),\tag{18}$$

with $\sigma$ the waist. Then for a fixed value of $N$, the equilibrium state corresponds to the value of $\sigma$ for which the energy functional Eq. (15) is minimized. We have verified that while for sufficiently large number of atoms there is a global minimum in $E(\sigma)$ at finite value of $\sigma$, corresponding to the droplet state, the later becomes a local minimum with $E(\sigma) > 0$ as one crosses the metastable point $N \leq N_{\text{meta}}$. Further decreasing $N$ one reaches the critical number $N_c$ below which the energy minimum at finite $\sigma$ vanishes and the ground state corresponds to a gas, $\sigma \to \infty$. The metastable number as well as the critical number of atoms as a function of $\delta g/g$ is reported in Fig. 6. We briefly note that within the variational approach, the

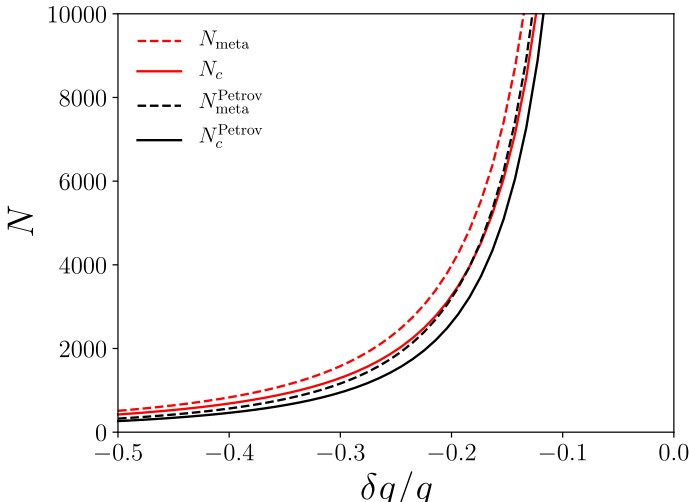

Figure 6: Critical number of atoms as a function of $\delta g/g$. The solid and dashed lines are the predicted atom numbers for the unstable ($N_c$) and metastable ($N_{\rm meta}$) droplet solution, respectively. The top red curves are evaluated within our theory, while the bottom black curves are calculated using the universal approach of Ref. [3] ($\tilde{N}_{\rm meta} = 24.03$ for the black dashed and $\tilde{N}_c = 19.61$ for the black dotted lines, respectively).

theory of Petrov predicts $\tilde{N}_{\rm meta} = 24.03$ and $\tilde{N}_c = 19.61$, thus slightly larger than the values $\tilde{N}_{\rm meta} = 22.55$ and $\tilde{N}_c = 18.65$ reported in Ref. [3], calculated from GPE (16). We find that the inclusion of beyond LHY terms in the energy functional is responsible for shifting the critical number to higher values. Experimentally, the critical number of atoms for the droplet state of Bose mixtures in both confined geometry [9] and free space [8] configuration has been measured. While in the free space measurement $N_c$ was found to lie near the prediction of Ref. [3] for the metastable state ($= 22.55$), the confined geometry measurement showed a deviation of $N_c$ to lower value. Recently, this deviation was accounted for the effects of finite-range in the interaction potential, which is neglected in the contact type $s$-wave description [12, 13].

## 5   1D Mixtures

Although the true Bose-Einstein condensation associated with an off-diagonal long-range order does not exist in one-dimensional geometry, it is known [35] that the Bogoliubov theory is quantitatively correct for predicting the energy in the regime where the coherence is sustained for distances large compared to the mean interparticle distance [36]. Thus, one can still use the Bogoliubov theory Eq. (2) to study the mixture, and the momentum sum in Eq. (3) can be evaluated straightforwardly, yielding the result

$$\frac{E}{V} = \frac{g}{2}\left(n_1^2 + n_2^2\right) + g_{12} n_1 n_2 - \frac{2}{3\pi}\frac{m^2}{\hbar}\left[c_d^3 + c_s^3\right], \tag{19}$$

with the interaction coupling constant related to the $s$-wave scattering length according to $g_{ij} = -2\hbar^2/(m a_{ij})$. It is worth noticing that in the 1D mixture, quantum fluctuations have opposite contribution (notice the negative sign) as compared to the 3D case Eq. (6). Consequently, droplets are formed in the dominantly repulsive regime [14] where $\delta g = g + g_{12} > 0$. Therefore the beyond mean-field energy functional (19) does not suffer from any complex

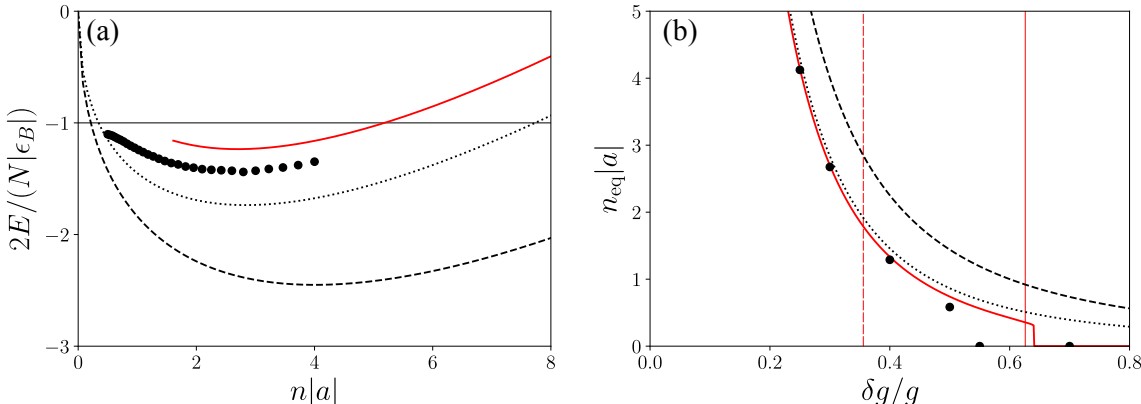

Figure 7: (a) Energy per particle of a 1D liquid as a function of density for $\delta g/g = 0.3$. (b) Equilibrium density of the liquid phase as a function of $\delta g/g$. The black dashed line corresponds to prescription (8) of Ref. [3] while the dotted line is the beyond mean-field result with the Bogoliubov speed of sound Eq. (7). The red solid line is the prediction from our theory. Black dots are the QMC results from Ref. [15]. The dashed and solid vertical lines in panel (b) indicate the critical values for $\delta g/g$ above which the liquid becomes metastable in respect to the molecular gas and atomic gas states, respectively.

sound velocities, in contrast to what happens in the 3D geometry. Nevertheless, it is instructive to obtain higher order corrections to the speed of sound, following a similar procedure as in the 3D case. After evaluating the compressibilities from Eqs. (5) and (19), we obtain the following expressions for the sound velocities:

$$c_d^2 = \frac{n}{2m}\left\{(g+g_{12}) - \frac{g}{2\pi}\frac{1}{\sqrt{n|a|}}\left[\left(1+\frac{g_{12}}{g}\right)^{3/2} + \left(1-\frac{g_{12}}{g}\right)^{3/2}\right]\right\} \tag{20a}$$

$$c_s^2 = \frac{n}{2m}(g-g_{12})\left\{1 - \frac{1}{\pi}\frac{1}{\sqrt{n|a|}}\frac{\delta g}{g_{12}}\left[\sqrt{1+\frac{g_{12}}{g}} - \sqrt{1-\frac{g_{12}}{g}}\right]\right\}. \tag{20b}$$

One can see that in 1D, higher-order corrections to the Bogoliubov speed of sound are responsible for a dynamic instability as $n|a| \to 0$. However, a peculiarity of 1D geometry is that the mean-field regime corresponds to the large density, $n|a| \gg 1$. In other words, the instability is predicted in the regime where the Bogoliubov theory is not applicable. It has been found in QMC simulations [15] that the liquid evaporates for $\delta g/g > 0.54$, in agreement with threshold value where effective interaction between dimers becomes attractive [37] while three-dimer interaction is still repulsive [38].

We show in Fig. 7(a) the equation of state of a one-dimensional Bose mixture for $\delta g/g = 0.3$ as predicted from our new theory and compared with QMC calculations from Ref. [15]. The energy is normalized by the binding energy of dimers composed from atoms from different components, $\epsilon_B = -\hbar^2/(ma_{12}^2)$. Finally, Fig. 7(b) shows the equilibrium density in the liquid phase, obtained from the minimum of the ground state energy. Surprisingly, we find that our theory essentially coincides with the QMC results up to extremely strong interactions, $\delta g/g \simeq 0.5$. By increasing $\delta g/g$, we find that the liquid phase becomes first metastable with respect to a molecular gas composed of $N/2$ free dimers, $E(n_{eq}) > N\epsilon_B/2$, with the threshold value $(\delta g/g)_{\text{Meta,1}} = 0.356$ indicated by the dashed vertical line in Fig. 7(b). Further increase in interactions make the liquid metastable in respect to the atomic gas state, $E(n_{eq}) > 0$, with the threshold value $(\delta g/g)_{\text{Meta,2}} = 0.626$ shown by a solid vertical line.

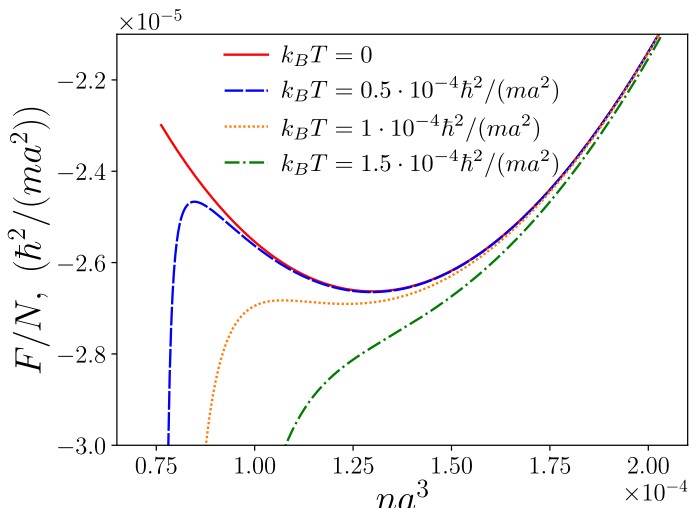

Figure 8: Free energy per particle of a 3D liquid Eq. (22) as a function of density for $\delta g/g = -0.2$ at different temperatures.

Eventually the minimum in the energy of the liquid disappears for $\delta g/g > 0.64$ making the liquid phase mechanically unstable. The exact threshold value is $\delta g/g > 0.54$ as found from QMC [15] and few-body calculations [37]. In other words, our predictions for the equilibrium density turn out to be very precise up to almost the threshold value where the liquid state disappears.

## 6 Effects of finite temperature

Finally, we discuss how the temperature affects the stability of the liquid phase. In the low temperature regime where $k_B T \ll |\mu|$, the thermodynamic behavior of a weakly interacting Bose gas is well described in terms of non-interacting phonons [4]. The Helmholtz free energy of the mixtures, both in three and one dimension, is therefore given by

$$F = E + k_B T \sum_{\mathbf{k}} \left[ \ln\left(1 - e^{-\beta \hbar c_d k}\right) + \ln\left(1 - e^{-\beta \hbar c_s k}\right) \right], \tag{21}$$

with $E$ the ground-state energy as given by Eq. (3).

Let us first discuss the 3D case. In a large system the sum over momenta in Eq. (21) can be approximated by an integral, and one finds the well-known $T^4$ law for the free energy:

$$\frac{F}{N} = \frac{E}{N} - \frac{\pi^2}{90} \frac{(k_B T)^4}{n\hbar^3} \left( \frac{1}{c_d^3} + \frac{1}{c_s^3} \right). \tag{22}$$

It is worth noticing that in the above expression, the speed of sound enters in the denominator, as $1/c_{d(s)}^3$. Thus for the description of thermodynamics, it is of fundamental interest to have a finite speed of sound, since approximating $c_d \simeq 0$ as in Eq. (8) would result in a droplet which becomes unstable at any small but finite temperature. We show in Fig. 8 the free energy of the mixtures calculated at different temperatures. For the chosen parameter $\delta g/g = -0.2$, we find that finite temperature has little effect as far as $k_B T \lesssim |E_{\text{eq}}|/N$, where $E_{\text{eq}}/N$ is the ground-state energy of the liquid, given by the minimum of the energy functional at $T = 0$. Increasing further the temperature, the liquid is predicted to evaporate at $k_B T_c \simeq 10^{-4}\hbar^2/(ma^2)$,

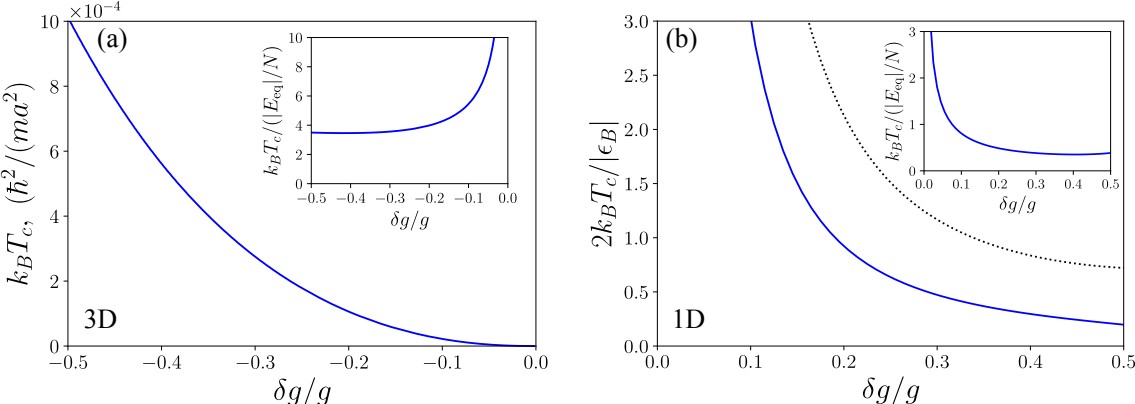

Figure 9: Critical temperature for the dynamical stability of the liquid phase, in (a) 3D and (b) 1D geometry. The inset shows the ratio of the critical temperature to the ground state energy of the liquid phase. In panel (b) the blue solid line is the prediction from our theory, whereas the black dotted line is the calculation using the Bogoliubov speed of sound Eq. (7).

slightly larger than the value $|E_{eq}|/N = -2.66 \cdot 10^{-5} \hbar^2/(ma^2)$. Figure 9(a) shows the estimation for the critical temperature $T_c$ in a 3D mixture as a function of $\delta g/g$. While the critical temperature decreases with decreasing $|\delta g|$, we find that in general $k_B T_c$ is of the same order as the ground-state energy of the liquid (see inset of Fig. 9(a)), apart from the region where $\delta g \to 0$. However, in this region, the critical temperature is found to be too large for applying the phonon thermodynamics, $k_B T_c \gg |\mu|$, and one needs to develop a finite-temperature theory which takes into account the excitation of single-particle states [39].

In the one-dimensional mixture, Eq. (21) for the free energy provides the result [40]:

$$\frac{F}{N} = \frac{E}{N} - \frac{\pi}{6} \frac{(k_B T)^2}{n\hbar} \left( \frac{1}{c_d} + \frac{1}{c_s} \right). \tag{23}$$

As we have already mentioned in Sec. 5, Bogoliubov theory in 1D geometry does not suffer from an imaginary speed of sound. We therefore show in Fig. 9(b) the calculated evaporation temperature, using both the beyond LHY approach Eq. (20) (blue solid line) and the Bogoliubov sound velocity Eq. (7) (black dotted line). We find that both approaches give the same qualitative behavior, with $T_c$ decreasing as one increases $\delta g$. This is the opposite behavior to the 3D case (see panel (a)) and it is understood as the fact that the ground-state energy of the liquid in 1D scales as $\propto 1/\delta g$ [14], in contrast to the 3D scaling $\propto |\delta g|^3$ (see Eq. (12)). Indeed, the evolution of the ratio $k_B T_c/(|E_{eq}|/N)$ shown in the inset of Fig. 9(b) confirms this picture, giving a behavior close to that of the 3D case. It is observed that for larger values of $|\delta g|$ the liquid becomes more unstable with respect to temperature as both quantum and thermal fluctuations destabilize the liquid. To summarize, we find that both in 3D and 1D geometry, the typical temperature leading to the instability of the liquid is given by the energy per particle (or the chemical potential) corresponding to the excitations of the soft mode, rather than dimer energy $\hbar^2/ma_{12}^2$ corresponding to the hard mode.

## 7 Conclusions

In conclusion, we have developed beyond-LHY theory for the description of self-bound quantum droplet in attractive mixtures of BECs. Our theoretical approach is based on self-consistent

inclusion of higher order terms in the sound velocities, calculated in a perturbative way. The corrections brought to the speed of sound in the density channel is shown to yield a real part, in contrast to the prediction of Bogoliubov theory which is purely imaginary, with a dramatic consequences in the thermal properties. The new sound velocities are used in turn to improve the energy functional. For the mixtures in three dimensions, our approach is found to describe accurately the equation of state in the liquid phase, predicting an equilibrium density in close agreement with available *ab-initio* calculations. We further investigate finite-size droplets by means of Gross-Pitaevskii equation, and calculate experimentally measurable quantities such as the size of the droplet and the critical number of atoms. As well, we construct beyond-Bogoliubov theory for 1D geometry. We find an excellent agreement with quantum Monte Carlo values for the equilibrium density while predictions of the Bogoliubov theory are rather imprecise. Finally, we study the thermal effects which are dominated by the excitations of the soft mode, for which our theory is needed to cure the imaginary values obtained in less accurate theories. We show that the minimum in the free energy disappears at temperatures of the order of the chemical potential thus making the liquid state dynamically unstable.

A natural extension of this work consists in investigating the asymmetric mixture. Indeed, on-going experiments use mixtures of $^{39}$K atoms in different hyperfine states, with $g_{11} \neq g_{12}$ [8, 9]. Recently the realization of self-bound Bose mixtures with different atomic component $m_1 \neq m_2$ has been also reported [41]. On the other hand, the theory developed in this work can also be extended for the dipolar [5, 6] and coherently coupled [42, 43] gases, which have attracted much interest these last years, due to the richness of its phase diagram [44–48].

*Note added.*-Recently, we became aware of related papers [49, 50] that examine the effects of bosonic pairing on the formation of self-bound quantum droplet, both at zero and finite temperature.

# Acknowledgements

We thank S. Giorgini, L. Parisi, S. Stringari and L. Tarruell for fruitful discussions.

**Funding information**   M. O. has received funding from the EU Horizon 2020 research and innovation programme under grant agreement No. 641122 QUIC, and by Provincia Autonoma di Trento. G. E. A. has been supported by the Ministerio de Economia, Industria y Competitividad (MINECO, Spain) under grant No. FIS2017-84114-C2-1-P and acknowledges financial support from Secretaria d'Universitats i Recerca del Departament d'Empresa i Coneixement de la Generalitat de Catalunya, co-funded by the European Union Regional Development Fund within the ERDF Operational Program of Catalunya (project QuantumCat, ref. 001-P-001644).

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
