# Peer review of "Beyond Lee-Huang-Yang description of self-bound Bose mixtures"

_SciPost Physics, doi:SciPost Phys. 9, 020 (2020)_

## Round 1 · Referee Report · Anonymous (Referee 1) · 2020-6-19

Report

The manuscript by M. Ota and G. Astrakharchik analyzes the properties of liquid droplets in 1D quantum Bose-Bose mixtures. The authors sensibly extend the available formalism by including corrections to the "standard" LHY treatment. These corrections are shown to have a number of very beneficial effects, and yield energies and equilibrium densities which are in excellent agreement with available QMC calculations. Moreover, the authors compute improved density profiles, droplet sizes and critical numbers of atoms for droplet formation which display a better agreement with existing experiments.

The manuscript is excellently written, the formalism is clearly outlined, and the results are thoroughly discussed. Moreover, this physics is certainly for ongoing experiments. As such, I can certainly recommend publication of this manuscript in SciPost.

Here below I collected a series of minor typos, remarks and comments (listed in chronological order) which the Authors may want to address upon resubmission.

Sec. 1

"The liquid droplets ... since THEY ARISE ..."

Sec. 2

appropriated --> appropriate

The authors mention here that corrections to the LHY theory may be obtained in two ways, following Beliaev theory, or through thermodynamics relations. Are the two results expected to match? or may one find different answers?

I see that this point is partially touched later, towards the end of Sec. III, but the authors may want to comment something here already. By the way, there may be a typo in Sec. III, because the authors write "a systematic calculation of the THIRD-order terms of the perturbative theory", but earlier Beliaev theory is introduced as being a SECOND order one. Please check?

atoms density --> atom density

disbalance --> imbalance

I couldn't find an explicit definition of the scattering length "a". It is obviously defined through "g", but the authors may want to add it, for completeness.

Sec. 3

Fig. 2: a black-dotted line is mentioned in the caption, and also in the main text, but I couldn't find it in the Figure.

Fig. 1 shows that the beyond-LHY correction to the spin mode is tiny in the liquid region. On the other hand, when discussing Fig. 2 the authors write that this has a strong effect on the energy. Is there a simple explanation for this? Or am I missing something? Please clarify.

Moreover, for easier reference in the caption of this figure and of Fig.7, I suggest to include the "equal density version" of Eq.(4) as a stand-alone equation right before Eq.(7).

Fig.3: the discrepancy between the improved theory and QMC seems to be maximal at $\delta g/g=-0.1$ (green), while it is much smaller at the next value shown, -0.05 (orange). Is there a simple reason for this?

caption of Fig.3 (And also text): "color dots" --> "colored dots"

"hints that that" --> "suggests that"

Sec. 4

the authors write "the main contributions to the LHY terms in Eq. (3) arise from “short” distances $\propto 1/c_+$". However, $c_+ \rightarrow 0$ in the region of interest. Could the authors clarify?

Eq.(14): is a $\Phi$ missing at the end of the equation? else, what is the Laplacian operator acting upon?

Eq.(16): if $\Phi$ is a 3D wavefunction, it should have dimensions of length^{-3/2}, but it seems to have length^{-3/4}; check the prefactor?

Fig.6: I'm confused: can one relate the values for $\tilde{N}$ given in the text with the data shown in this figure? For which value of $n_0\xi^3$ is the Figure obtained? (or is this not relevant?) Please clarify.

Sec.6:

do the corrections to the speed of sound yield a "real part"? or rather a "real result"?

could the results found here be used to evaluate finite-temperature thermodynamics of such binary mixtures?

  • validity: top
  • significance: high
  • originality: top
  • clarity: top
  • formatting: perfect
  • grammar: excellent

Author:  Miki Ota  on 2020-07-29  [id 910]

(in reply to Report 1 on 2020-06-19)
Category:
answer to question

We thank the Referee for the insightful comments and for the useful suggestions for improvement. Below we provide detailed answers to the comments:

Sec. 2. The authors mention here that corrections to the LHY theory may be obtained in two ways, following Beliaev theory, or through thermodynamics relations. Are the two results expected to match? or may one find different answers? I see that this point is partially touched later, towards the end of Sec. III, but the authors may want to comment something here already.

The Beliaev theory is known to yield the same ground-state energy as the LHY theory, and the main difference is that the single-particle excitation spectrum is improved in respect to the Bogoliubov theory. In particular, in the long-wavelength limit, the phonon mode acquires both real and imaginary additional terms, the later being known as the Beliaev damping. It is this real part correction, which we calculate macroscopically through the evaluation of the compressibilities. The results found from our approach and the diagrammatic method of Beliaev are expected to match, exactly for the compressibility (and the speed of sound of the density mode) while we are missing additional corrections for the magnetic susceptibility (i.e. speed of sound of the spin mode). This is because in our approach we do not take into account the microscopic Andreev-Bashkin (AB) effects, which arise from entailment between two superfluids. However, for the weakly interacting Bose mixtures, the AB corrections to the spin susceptibility was found to be negligible [24] with respect to the beyond mean-field correction found in our work, therefore supporting our approach and suggesting its good accuracy. The final verification is done by testing predictions of our theory against Quantum Monte Carlo data and a very good agreement is found. We also note that the identification between the microscopic phonon and the macroscopic sound does not hold anymore for an asymmetric mixture, and one needs to rely on the Beliaev theory in order to calculate the speed of sound. We will add a paragraph explaining this point before Eq. (5).

By the way, there may be a typo in Sec. III, because the authors write "a systematic calculation of the THIRD-order terms of the perturbative theory", but earlier Beliaev theory is introduced as being a SECOND order one. Please check?

In the mentioned phrase, the differences in the energy between our theory and Monte Carlo results are attributed to the third-order terms for which we do not have a complete theory. The Beliaev and Bogoliubov theories are second order ones and provide the same prediction for the energy. Our approach, based on using thermodynamic relations to correct the speed of sound and evaluate the additional contributions to the energy, generates some of the third-order terms, but not all of them.

Sec. 2 I couldn't find an explicit definition of the scattering length "a". It is obviously defined through "g", but the authors may want to add it, for completeness.

We will add the definition of the coupling constant in terms of the scattering length below Eq. (1).

Sec. 3 Fig. 2: a black-dotted line is mentioned in the caption, and also in the main text, but I couldn't find it in the Figure.

We will add the missing curve in the new version of the paper.

Sec. 3 Fig. 1 shows that the beyond-LHY correction to the spin mode is tiny in the liquid region. On the other hand, when discussing Fig. 2 the authors write that this has a strong effect on the energy. Is there a simple explanation for this? Or am I missing something? Please clarify.

Indeed, the ultradilute liquids differ from usual single component gases in that the importance of beyond mean-field terms is drastically enhanced. The threshold for the formation of the liquid such that the mean-field repulsive and attractive terms cancel each other exactly. Thus, the importance of the subsequent terms is greatly enhanced. The present system is rather peculiar as conceptually it provides an “easy” experimental access to beyond-LHY terms while for single component gases the experimental measurement even of LHY terms is already quite complicated.

More formally, one can express the energy functional in terms of the universal equilibrium energy $E_0$ and density $n_0$ given by Eq. (12) and (13), and expand it in series of small parameter $\delta g / g$. One finds:

$$ \frac{E}{|E_0|} \simeq -3\frac{n}{n_0} + 2 \left(\frac{n}{n_0}\right)^{3/2} + \frac{5}{2}\frac{|\delta g|}{g} \left(\frac{n}{n_0}\right)^{3/2} - \frac{5}{4} \left(\frac{\delta g}{g}\right)^2 \left(\frac{n}{n_0}\right)^{3/2}\left(\frac{5}{3}\sqrt{\frac{n}{n_0}} - \frac{3}{2}\right) . $$
The two first terms are identified as the ones of the theory of Petrov, and expressed in these units do not depend explicitly on $\delta g$. The third term comes from the spin sound of the Bogoliubov theory, and gives a positive shift, as one can verify on Fig. 2. Finally, the last terms come from the corrections brought to the spin sound within our new theory. That is the mentioned corrections arise from the correction in the spin sound. Instead, the first contribution of the density sound to the energy functional enters with a higher power as $(\vert \delta g \vert / g)^{5/2}$. Even though the speed of density sound is drastically modified in our theory, its effects to the energy remains therefore tiny.

We will add the above formal analysis in the main text too, in Sec. 3.

Moreover, for easier reference in the caption of this figure and of Fig.7, I suggest to include the "equal density version" of Eq.(4) as a stand-alone equation right before Eq.(7).

We will add an explicit equation for the speed of sound in the equal density case, new Eq. (7).

Sec. 3 Fig. 3: the discrepancy between the improved theory and QMC seems to be maximal at $\delta g/g=−0.1$ (green), while it is much smaller at the next value shown, -0.05 (orange). Is there a simple reason for this?

We have carried out a systematic study of the discrepancy between our theory and QMC, which you can find on the attached figure "E_comparison.pdf". We have actually found different behavior for the discrepancy $|(E-E^\mathrm{QMC})/E_0|$ depending on the density of the liquid. For density larger than the typical equilibrium density $n/n_0 \ll 1$, we find that the error is larger for larger value of $|\delta g|/g$, which is an expected behavior since our theory is perturbative in $|\delta g|/g$. However, for smaller values of density $n/n_0 < 1$ we find that the error becomes maximal for $\delta g / g = -0.1$ as pointed out by the referee. We could not find any simple reason behind the observed behavior, which may arise for instance from the finite-range effects present in QMC simulation due to the used finite-size potentials. Finally, the value of $|(E-E^\mathrm{QMC})/E_0|$ decreasing as one decreases $n/n_0$ reflects the fact that perturbative theory works better at smaller density.

Sec. 4 # the authors write "the main contributions to the LHY terms in Eq. (3) arise from “short” distances $\propto 1/c_+$ ". However, $c_+ \to 0$ in the region of interest. Could the authors clarify?

There was indeed an inconsistency concerning the notation for the speed of sound $c_\pm$, between the definition given for the Bogoliubov sound Eq. (4) referring to the upper ($+$) and lower ($-$) branch of the excitation spectrum, and the definition used in the remaining of the paper. This was at the origin of the inconsistency pointed out by the referee. In the new version of the paper, we introduce the notation $c_d(s)$ for the density ($c_d$) and spin ($c_s$) mode, related to the compressibility (now defined as $\kappa_d$) and spin susceptibility (now defined as $\kappa_s$), respectively. Under this new definition, $c_s$ is related to the short distance physics.

Sec. 4 # Eq.(14): is a $\Phi$ missing at the end of the equation? else, what is the Laplacian operator acting upon?

There was a missing $\Phi$ in the equation indeed.

Sec. 4 # Eq.(16): if $\Phi$ is a 3D wavefunction, it should have dimensions of length^{-3/2}, but it seems to have length^{-3/4}; check the prefactor?

For the normalization, we have chosen $\int \vert \Phi \vert^2 = N / n_0$ (see after Eq. (14)) so that its dimension is correct.

Sec. 4 Fig.6: I'm confused: can one relate the values for $\tilde{N}$ given in the text with the data shown in this figure? For which value of $n_0 \xi^3$ is the Figure obtained? (or is this not relevant?) Please clarify.

Indeed, it is possible to relate the values of $\tilde{N}$ to the curves shown in Fig. 6, at least for the Petrov theory. In our approach, the critical $\tilde{N}$ will depend on the value of $\delta g$, so that we can not attribute the plotted curve to a single value of $\tilde{N}$. We will add the corresponding values for the Petrov theory in the caption. Here $n_0 \xi^3$ is not relevant, since it is a function of $\delta g / g$, used for the x-axis. We report now the values of $\tilde{N}$ for the Petrov theory in the caption of the figure.

Sec. 6 # do the corrections to the speed of sound yield a "real part"? or rather a "real result"?

It yields a real part, the speed of sound being still complex. See comments to Referee 2 for further details.

Sec. 6 # could the results found here be used to evaluate finite-temperature thermodynamics of such binary mixtures?

The presented approach can indeed be extended at finite-temperature, in a straightforward way as concerning the low temperature region where one can safely consider a gas of non-interacting phonons. This question of the referee motivated us to add a new section in the new version of the paper, where we calculate the Helmholtz free energy of the mixture in the phonon regime, and discuss the outcome. Interestingly, we find that the liquid becomes unstable when thermal fluctuations are taken into account, and evaporates at a critical temperature which is found to be of the same order as the equilibrium energy of the bulk liquid.

Attachment:

E_comparison.pdf

---

## Round 1 · Referee Report · Alberto Cappellaro (Referee 2) · 2020-6-20

Strengths

1 - A much needed analysis finally addressing a "loophole" in the original model for self-bound quantum droplets arising in binary Bose mixtures.

2 - The key idea is very transparent and relies upon a very simple argument, avoiding unnecessary complications.

3 - The comparison between the analytical predictions and QMC numerical outcomes reinforces the strength of the analysis and the assumptions on which it is built on.

Weaknesses

  • Despite repeatedly mentioning the availability of experimental data, the paper somewhat lacks a satisfactory comparison between theory and experiments.

  • Connected to the point above, since the approach is any case perturbative, it is not clear if the detachments from Petrov's original proposal predicted here may actually be revealed.

Report

The manuscript by Ota and Astrakharchik is focused on the solution of a known issue in Petrov's seminal proposal about quantum mechanical stabilization of binary Bose mixtures. Indeed, it is a known fact that one branch of the collective excitation spectrum (the density one) becomes purely imaginary beyond the mean-field instability threshold.

By computing beyond-Gaussian corrections to the Bogoliubov sound, the authors manage to extract analytical meaningful result, hopefully better fitting the increasing amount of data coming from current experimental setups.

In my opinion, this investigation for sure deserves the publication in SciPost, but I'd like the authors to address some minor points I am listing below:

1) The authors notice that both the density and spin compressibility in Eqs. (8)-(9) display a non-zero imaginary part. They then mention the fact that this is an expected outcome for a perturbative approach and that it can be safely neglected within the current experimental regimes. I'd like the authors to expand a little bit on why it is a safe procedure, since the reader may have the impression that the pitfall of Petrov's theory (the appearance of imaginary quantities signalling some sort of instability) is simply shifted towards the compressibility.

2) At page 6, just below the caption of Fig. 2, the authors refer to the spinodal point pointing out the instability of the liquid phase towards the nucleation of multiple droplets. Has this process ever been observed? It seems to me a very relevant experimental point and a comment should be in order.

3) In the caption of Fig. 2 the authors mention five different lines, but I do not see the black dotted one, neither in the figure or in the legend.

4) Again about Fig. 2 and its discussion in page 7, I certainly agree with the claim that the main correction to the energy density comes from the spin channel, but I do not understand what is the strong suppression of energy one should see comparing the green and the blue line. I think the authors should clarify this point.

5) Concerning Eq. (13), it seems important to mention that such an effective description with only a single field $\Phi$ actually neglects the eventual internal dynamics between the mixture components.

6) In page 10-11, while discussing the results of their GPE calculations based on the effective energy functional in Eq. (10), the author often mention relevant experimental investigations about the self-bound droplets such as Refs. (8-10), but I do not see a clear comparison between their theory and such investigations. Is it because the experimental setups are still lacking the sensitivity to probe this detachment from Petrov's theory? If not, I think this work would greatly benefit from a more extended discussion (maybe a figure, if data are available) about this point.

Related to the point above, at the end of Section 4 (page 11), the authors writes about "trap confined measurements" but do not cite any reference.

7) In the conclusion, the authors mention the possible extension of their theory, I would also add the possibility to deal with a Rabi-coupled mixtures and also a spin-orbit coupling. In both cases, theoretical investigations base on Petrov's theory are already available.

  • validity: high
  • significance: good
  • originality: good
  • clarity: high
  • formatting: excellent
  • grammar: excellent

Author:  Miki Ota  on 2020-07-29  [id 911]

(in reply to Report 2 by Alberto Cappellaro on 2020-06-20)
Category:
answer to question

We thank the Referee for the insightful comments and for the useful suggestions for improvement. First, the Referee is concerned about the lack of a comparison with experiments. We have not included direct comparison between theory and experiment because available experiments are carried for asymmetric mixtures only, and in this work we have focused on symmetric configuration. The theory is simpler and more elegant in the symmetric case. Nevertheless, experiments have found discrepancies with Petrov approach, which go along with the results found in our theory. The extension of the present theory to the asymmetric case for a direct comparison with these experiments is one of our future directions of the work.

Below are the answers to the specific questions:

1) The authors notice that both the density and spin compressibility in Eqs. (8)-(9) display a non-zero imaginary part. They then mention the fact that this is an expected outcome for a perturbative approach and that it can be safely neglected within the current experimental regimes. I'd like the authors to expand a little bit on why it is a safe procedure, since the reader may have the impression that the pitfall of Petrov's theory (the appearance of imaginary quantities signalling some sort of instability) is simply shifted towards the compressibility.

It is indeed true that the velocity of sound remains complex in our approach, as a result of complex compressibilities. This can not be avoided as long as one starts from the LHY energy functional with the Bogoliubov speed of sound entering in. However, the crucial difference with the Petrov approach stems from the fact that in our theory the sound velocity in the density channel yields a large real part and small imaginary one instead of entirely imaginary value in the Petrov theory. In fact, one finds from Eq. (9) that the imaginary part of the compressibility is smaller than the real part by a magnitude $\vert \delta g \vert^{5/2}$. Since in available experiments $\vert \delta g \vert \ll 1$, one finds that the imaginary part is negligible in respect to the real part. Finally, in an analogous way to what we have found, one can expect that third order perturbation theory will further shift the imaginary part to higher order terms, and so on, so that the imaginary part of the compressibility eventually vanishes in a non-perturbative approach. A short discussion to better clarify this point will be added in Sec. 2.

2) At page 6, just below the caption of Fig. 2, the authors refer to the spinodal point pointing out the instability of the liquid phase towards the nucleation of multiple droplets. Has this process ever been observed? It seems to me a very relevant experimental point and a comment should be in order.

This effect is entirely analogous to the fragmentation of classical liquids at large enough negative pressures. In quantum ultradilude liquids this instability has been observed in Monte Carlo simulations. Experimentally, production of multiple droplets have been observed, when the ramp of the magnetic field to tune the interaction or the removing of the trapping potential was not achieved adiabatically. However, it is difficult to associate it uniquely to the observation of the spinodal point. A direct way to observe it would be to apply negative pressure, or to start with an almost uniform liquid and quench the interactions to such parameters that the uniform system will be unstable. We have added a sentence discussing this point at the end of Sec. 2.

3) In the caption of Fig. 2 the authors mention five different lines, but I do not see the black dotted one, neither in the figure or in the legend.

A curve in Fig. 2 was indeed missing. We add it in the new version of the manuscript.

4) Again about Fig. 2 and its discussion in page 7, I certainly agree with the claim that the main correction to the energy density comes from the spin channel, but I do not understand what is the strong suppression of energy one should see comparing the green and the blue line. I think the authors should clarify this point.

The suppression of energy observed in Fig. 2 is best understood if we express the energy functional in terms of the equilibrium energy $E_0$ and density $n_0$ given by Eq. (12) and (13), and expand it in series of $\delta g / g$:

$$ \frac{E}{|E_0|} \simeq -3\frac{n}{n_0} + 2 \left(\frac{n}{n_0}\right)^{3/2} + \frac{5}{2}\frac{|\delta g|}{g} \left(\frac{n}{n_0}\right)^{3/2} - \frac{5}{4} \left(\frac{\delta g}{g}\right)^2 \left(\frac{n}{n_0}\right)^{3/2}\left(\frac{5}{3}\sqrt{\frac{n}{n_0}} - \frac{3}{2}\right) . $$
The two first terms are identified as the ones of the theory of Petrov, and do not depend explicitly on $\delta g$. The third term comes from the spin sound of the Bogoliubov theory, and gives a positive shift, as one can verify on Fig. 2. Finally, the last terms come from the corrections brought to the spin sound within our new theory. It is a negative contribution in the region where $n/n_0 \gtrsim 1$, resulting in a suppression of the energy. Because of its density dependence, $\propto(n/n_0)^{3/2}$, this suppression becomes larger as the density increases. As for the contribution from the density mode, its first contribution to the energy functional enters as at a higher power of the small parameter, $(\vert \delta g \vert / g)^{5/2}$. Although the speed of density sound is drastically modified in our theory, its effect to the energy remains therefore tiny.

We will add the above formal analysis in the main text too, in Sec. 3.

5) Concerning Eq. (13), it seems important to mention that such an effective description with only a single field Φ actually neglects the eventual internal dynamics between the mixture components.

We will add a corresponding clarification in the new version.

6) In page 10-11, while discussing the results of their GPE calculations based on the effective energy functional in Eq. (10), the author often mention relevant experimental investigations about the self-bound droplets such as Refs. (8-10), but I do not see a clear comparison between their theory and such investigations. Is it because the experimental setups are still lacking the sensitivity to probe this detachment from Petrov's theory? If not, I think this work would greatly benefit from a more extended discussion (maybe a figure, if data are available) about this point. Related to the point above, at the end of Section 4 (page 11), the authors write about "trap confined measurements" but do not cite any reference.

As we have already mentioned at the beginning of this report, the lack of comparison with experimental data comes from the assumed simplification of symmetric interactions (repulsion in component one is exactly the same as in component two). Unfortunately, the theory becomes cumbersome in the asymmetric case, while in experiments it was not yet possible to create a liquid with symmetric interactions. One can nevertheless compare our theory with available experiments using $^{39}K$ if we take as reference the prediction of Petrov theory, and compare the data for the same value of $\delta g / g = 1 - g_{12} / \sqrt{g_{11} g_{22}}$. In the experiments, $\delta g / g < -0.15$, and for the largest value of $\delta g / g$ Ref. [10] finds for the critical number of atoms a good agreement with Petrov theory within the errorbars. For the same value of $\delta g / g$, our theory predicts a correction to the Petrov theory of about $18\%$ (see Fig. 6), which also lies within the errorbars of the experiment. In fact, the experiment finds a slightly higher value of $N_c$ in respect to Petrov, thus closer to our prediction. But since the experimental errorbars are big, currently it is not possible to discern the two predictions in the critical number. Therefore, a better sensitivity is needed in the experiment, or measurements for larger values of $\vert g_{12} \vert$.

We will add a discussion about the experimental relevance of our predictions at the end of Sec. 2. Of particular interest is the possibility of measuring speed of density and spin modes as this can be done with the available high-precision experimental techniques. Furthermore we provide a qualitative effect which as well could be experimentally observed, consisting in the fragmentation instability for the densities smaller than the spinodal one.

7) In the conclusion, the authors mention the possible extension of their theory, I would also add the possibility to deal with a Rabi-coupled mixtures and also a spin-orbit coupling. In both cases, theoretical investigations based on Petrov's theory are already available.

Indeed, that can be a very interesting direction of investigation. We will mention it in Conclusions.

---

## Round 2 · Author Response

Dear Editor,

we want to thank you for handling the manuscript. We are also grateful to the Referees for their positive feedbacks about our work, as well as for their valuable comments.

We would like to submit a new version of our paper, taking into account the suggestions made by the Referees. Below we give the details about the modifications brought in this new version.
We hope that our manuscript will be accepted soon for publication in SciPost Physics.

Sincerely Yours,

Miki Ota,
Grigori E. Astrakharchik

---

## Round 2 · List of Changes

# General changes
- We have added a new section (Sec. 6) discussing the stability of the droplet at finite-temperature.
- We have changed the notation for the speed of sound, as well as the compressibilities, for more clarity. We now denote by $c_d$ the density sound, and $c_s$ the spin sound, and avoid the notation $\pm$ in the whole paper.

# Regarding the requests from Referee 1

- Sec. 2 p3 We have added the definition of the coupling constant in terms of the *s*-wave scattering length, after Eq. (1).
- Sec. 2 p4 We have added a discussion about the relationship between our macroscopic approach and the Beliaev diagrammatic theory, before Eq. (5).
- Sec. 3 p5 We have added the expression for the Bogoliubov sound in the symmetric case Eq. (7).
- Sec. 3 p7 We have added the missing curve (black dotted one) in Fig. 2.
- Sec. 3 p7 We have added a formal discussion about the contribution of each theory on the energy functional, Eq. (14).
- Sec. 4 p10 We have added the missing $\Phi$ on the right-hand side of Eq. (16).
- Sec. 4 p12 We have added the values of $\tilde{N}$ for the Petrov approach in the caption of Fig. 6.

# Regarding the requests from Referee 2
- Sec. 2 p5 We have added a sentence clarifying the origin of complex value in the compressibility, before Eq. (11).
- Sec. 2 p6 We have added a discussion about the experimental possibility to observe the theoretical findings made in of our work.
- Sec. 3 p7 We have added the missing curve (black dotted one) in Fig. 2.
- Sec. 3 p7 We have added a formal discussion about the contribution of each theory on the energy functional, Eq. (14).
- Sec. 4 p10 We have added a sentence about the limitation of the single-mode approximation $\Phi_1 = \Phi_2$, after Eq. (15).
- Sec. 7 p16 We have added a sentence about coherent coupled BECs as an open-question.

# Other minor changes
- We have added a sentence in the abstract, as well as in the introduction, referring to the newly added finite-temperature study.
- We have fixed an error found in the calculation of the black dotted line in Fig. 7(b).
- We have added a note at the end of the manuscript to refer to recent works on the same topic appeared recently as preprints.
- We have fixed some typography misses.

---

## Editorial Decision

published